# Gamification and Controversial Heritage: Trainee Teachers' Conceptions

Sergio Sampedro-Martín *, Elisa Arroyo-Mora , José María Cuenca-López and Myriam José Martín-Cáceres

Integrated Didactics Department, Faculty of Education, Psychology and CC. Sport, University of Huelva, 21007 Huelva, Spain; elisa.arroyo@ddi.uhu.es (E.A.-M.); jcuenca@ddcc.uhu.es (J.M.C.-L.); myriam.martin@ddcc.uhu.es (M.J.M.-C.)
* Correspondence: sergio.sampedro@ddi.uhu.es; Tel.: +34-6-453-02411

**Abstract:** Heritage education is configured as an ideal framework for the treatment of socio-environmental problems relevant to modern society. To this end, it is essential that teachers can develop proposals in the classroom that promote critical thinking and the eco-social education of their students, based on innovative and experiential methodologies. For this reason, initial teacher training must ensure that future teachers acquire these innovative tools. Thus, the aim of this work is to identify the conceptions of student teachers in initial teacher training regarding gamification as a teaching methodology to address controversial heritage. To achieve this, a questionnaire has been designed with questions about heritage, controversy, and educational gamification, which was given to 235 students (M = 60; F = 175) taking their primary education degrees at different Spanish universities. To guide the design of the research instrument and analyze the results, a system of categories was developed and the SPSS data processing program was used. The first results show that, even though students in initial teacher training think that they do have a predisposition to use gamification to work around controversial heritage in class, they lack the knowledge of methodological strategies and gamified educational activities, which suggests a dearth of training on these issues in the primary education degree courses.

**Keywords:** heritage education; gamification; controversial heritage; teacher training; quantitative research; eco-social education

## 1. Introduction

The challenge of teaching social knowledge that is useful for students and that will enable them to function with autonomy, responsibility, and critical capacity in today's societies [1] entails a continuous process of seeking new and alternative forms of teaching so that teachers undergoing initial training can learn active methodologies that are capable of generating more significant learning in their future students and educating them as critical citizens who will be involved in tackling sustainable development and today's most relevant eco-social issues [2].

Along the same lines, the EPITEC2 project proposes the incorporation of three perspectives of enormous educational potential: relevant socio-environmental problems (hereinafter, RSPs), the ecosocial approach, and controversial issues [3]. Heritage education (hereinafter referred to as HE) is a key element of comprehensive training, from the earliest stages of education to the training of the teachers themselves [4–6], based on the so-called controversial heritage. In this sense, HE promotes the development of critical thinking and the related reflection processes for the formation of a democratic, participatory, transformative, and just eco-citizenship [7]. To this end, models of future teacher education that are based on the realist perspective and school-based research are successfully employed, with educators seeking to ensure the integration of personal experiences in classroom practice alongside purely theoretical knowledge [8]. This realistic model and the didactic tools and strategies that are brought into play contribute to the reconstruction of previous experiences

and beliefs that could lead to a negative professional identity [9]. This is a fundamental aspect of initial teacher training since, in many cases, after having completed their initial university training, new teachers enter the profession and reproduce the same models that they experienced during their pre-university training [10]. It is, therefore, necessary to continue researching didactic instruments and strategies that favor the joint construction and reconstruction of knowledge in all phases of initial teacher training [11].

HE is configured as the ideal framework for addressing the relevant socio-environmental issues of our society [12]; to this end, it is essential that teachers know how to develop proposals in the classroom that promote critical thinking in their students, based on innovative and experiential methodologies [13]. Thus, gamification, understood as the use of game design elements in non-game contexts [14], may be one of the key ideas that help to improve the working mechanisms of initial teacher training since it makes it possible to bring the course content closer and develop the competencies of university students, especially those related to education [15].

### 1.1. Controversy: Relevant Socio-Environmental Issues and Controversial Heritages

Over the last century, through socially current issues [16] and critical theory [17], controversial themes have been introduced into the teaching and learning process in order to overcome the uncritical transmission and reception of memorized and partial content and to promote the reflective and critical spirit of students [2].

According to Kerr and Huddleston [18], it is necessary to include these controversial issues in the classroom because of their importance in society, the debate that is generated as part of the democratic process, student participation in the context of receiving and sharing information, the continuous emergence of new controversies, the analysis of controversy by putting critical and analytical thinking skills into practice, the inclusion of highly relevant current issues, or because of the introduction of controversial topics in the classroom by students, among other causes. These same authors state that controversial topics that can be introduced in the classroom can be classified into established topics and very recent subjects, according to their temporality, which will condition the relevance that teachers and students attribute to these themes. Similarly, Stradling [19] distinguishes between those issues that can be resolved by evidence, the controversy of which is therefore superficial, and those where the dispute is inherently disputed, i.e., issues arising from disagreements based on matters of fundamental belief or value judgments. At this point, we can also differentiate between controversial issues according to the context they affect, so that we distinguish between local and global issues.

The Council of Europe advocates "addressing controversial issues for the education of a critical citizenship, which develops a socially transformative democratic engagement" [3] (p. 484), while stressing, through the principles set out in the 2005 Faro Convention, the importance of HE in terms of socially contentious issues. Thus, in the Spanish context, the EPITEC Project2 advocates working on RSPs on the basis of controversial heritage, which is defined as perspectives on those heritage elements that are didactically selected in response to various causes that give rise to or generate conflict, controversy, dilemma, or debate [7]. The educational aim of addressing these controversial heritage issues with future teachers is:

> For students in initial teacher training to analyze hegemonic history critically, along with the hierarchization of relationships, human domination over bodies and territories and the discrimination, marginalization, and/or oppression of certain social groups and be able to take individual and collective action to build a more just, peaceful, egalitarian, and sustainable society [20] (p. 69).

The opportunity we find here lies in the fact that the teaching of these heritage perspectives allows us to work on these controversial issues from an eco-social approach, in order to form a citizenry committed to advocating and participating in the management, conservation, and safeguarding of heritage [21,22]. In this way, it will be possible to achieve an HE that enables the acquisition of a greater commitment to their community,

understanding, and reflection on the possible consequences of the acts that have occurred between past, present, and future connections, in order to understand and value them from a critical and constructive perspective [23,24].

Heritage is, therefore, an educational resource that awakens students' motivation, makes school content useful for the socio-environmental transformation of their environment, and promotes meaningful or deep learning [25]. In this way, RSPs and controversial heritage appear as suitable elements for work in initial teacher training, as teaching content, or as resources for the development of future teachers, building an eco-citizenship that seeks social transformation [20].

### 1.2. Gamification: Innovative Methodologies in Initial Teacher Training

In line with the formation of an active and reflective citizenry committed to change and social justice [26], initial teacher training is a key aspect of ensuring that these education professionals should be suitably trained to carry out the design and experimentation of teaching materials and proposals in this field [27].

Based on the assumption that controversial heritage serves to address RSPs in the classroom as well as the eco-social education of citizens, and that, in turn, school research on these topics favors the development of critical thinking, we should focus on the conceptions that student teachers in initial training already have of teaching processes [12]. The aim is to clarify whether future teachers have the necessary skills to be able to act regarding these issues and, therefore, whether they need training in the appropriate handling of controversial issues to encourage the development of critical thinking [3].

In education, new techniques related to gamification are increasingly being implemented because they exhibit particularly interesting benefits for both students and teachers [28–30]. In gamified activities, relationships are established between the environment, the players, and each other as they express their emotions, gain experiences, have fun, relax, and find solutions to problems [31]. Moreover, learners show positive attitudes towards the gamification of content, as it encourages student motivation and, in general, helps them to receive positive feedback during the teaching process [32]. It can be affirmed that gamified activities help to improve communication skills and also coordination and collaboration between peers, improving the inclusion in the classroom of all participants in the game. In addition, they promote a positive attitude in students regarding the achievement of academic success [31].

According to a study by Perrota et al. [33], the use of gamification in the classroom appears to improve learning outcomes and increase student engagement with educational processes. Active and gamified methodologies of this kind generate a high level of motivation in university students, allow cooperative work, and favor the acquisition of key competencies and subject contents [34].

Based on the scientific literature, we find several precedents on gamification in initial teacher education [35–39], although it is only on rare occasions in these precedents that any gamified activity is used to work on current controversial topics, such as in the studies by Clarke et al. [40] or Nicholson [28]. However, pre-service and in-service teachers still show a high level of ignorance about the educational use of gamified strategies [41,42]. In addition, no previous work linking controversial issues, gamification, and heritage in initial teacher education was found. This means that teachers do not find adequate references to unify the potential of gamification with the benefits of controversy-based HE, thus hindering the dissemination and use of these types of strategies in the classroom to address current RSPs. It is, therefore, essential to ensure that future teachers are trained in strategies that favor a holistic perspective when dealing with knowledge, beliefs, and feelings that facilitate the beginning of the construction of critical thinking and increase motivation, inclusion, and group cohesion, and that are considered to be optimal alternative methodological strategies that place students at the center of their own learning in an active and cooperative way [28,29].

One example of a gamified activity is role-playing games, which consist of a collective game composed of a storyteller and participants who adopt roles and create or recreate stories [43]. The role-playing game allows an empathic knowledge of reality, while its structure entails a certain amount of group work, enabling interaction between equals, and involving creative situations that students interpret as a means of knowledge acquisition [10]. Roda [44] distinguishes between four modalities of role-playing (live role-play, written role-play, table role-play, and role-playing video games) with which to combine multiple options for implementing this strategy in the classroom. Role-playing games represent an innovative didactic proposal in the teaching of social sciences, offering multiple educational advantages when addressing relevant problems, helping students to develop informed opinions, stimulate critical thinking, develop social and argumentation skills, and emphasize the procedural and attitudinal aspects [42].

The following strategies share similarities with each other, mainly in that they are both fun, motivating, and exciting elements, where respect and collaboration are maintained while developing communication skills and critical thinking [45], and they can both be applied relatively easily in class. Firstly, the 'Escape Room' is a game in which a group of people has to collaborate actively in the resolution of different enigmas and problems to achieve the aim of getting out of a room [31,46,47]. Conversely, 'Breakout' consists of opening a closed box with different types of locks; it is necessary to solve problems to find the codes that open them [48]. Several experts, such as Nicholson [28] or Veldkamp et al. [30], are introducing these strategies because they offer great benefits that are of interest to both students and teachers. According to García-Lázaro [29], the peculiarity of Escape Rooms is that they are a resource that demands the inclusion of a great variety of personalities in the players. In turn, Moreno-Fuentes [34] stated that Breakout is an experience that is more closely related to educational action.

Another example of gamified activity is 'Civil Dialog', a structured, simulated debating technique for controversial issues, in which participants are asked to take sides by sitting in chairs arranged in a semicircle and presenting their ideas, which range from 'Strongly agree' to 'Strongly disagree' [49]. The format was created in 2004 as a way to explore citizens' reactions to political rhetoric. It was developed at the Hugh Downs School of Human Communication at Arizona State University and has continued to be developed by John Genette, Jennifer Linde, and Clark Olson, among other academics. This technique provides a venue for civilized and facilitated citizen dialogs on the issues of our time [50]. A Civil Dialog round is composed of a moderator (teacher), the audience (student body), and 5 participants from that audience and usually lasts between 30 and 45 min. This activity is less well known, but it represents a great tool with enormous potential for working on social problems of current relevance and fosters positive values, such as empathy and respect, by being able to place each student in positions contrary to their own if combined with roleplay, as can be seen in the study by Cruz-Lorite et al. [39].

Based on the experiences found, the range of gamified activities that can be used in classrooms is very wide. Using these prior studies as a starting point, we can establish a classification of those gamified activities that may be most useful for the teaching of controversial heritage, based on the following features:

- Static role-playing games: this includes written role-playing games, board games, and similar games.
- Videogames: all games that need some kind of technological support in order to be played (mobile, tablet, computer, console, etc.).
- Breakouts: enigmas, puzzles, and brainteasers that can be solved without leaving the classroom.
- Escape Rooms: enigmas, puzzles, and brain teasers that need to be solved to move from one space to another.
- Historical re-enactments: activity in which participants recreate some aspects of a historical event or period. This may be narrowly defined, such as a war or other specific event, or have broader coverage.

- Live role-play: a first-person game in which the representation of the characters by the players takes place in real time and in a staged manner.
- Civil Dialogs: simulation of debate on current conflicts, adding the use of role-playing to assign divergent positions.

## 2. Materials and Methods

The research presented here is an exploratory study [51,52], which is designed to address the teaching of controversial heritages through gamified methodological strategies in initial teacher training, an issue that has yet to be explored in the context of HE. To this end, this paper attempts to highlight methodological issues, detect limitations of the research and show the validity and reliability of the instruments applied.

Moreover, this study follows a multiple paradigmatic approach as its design is based on an interpretative or naturalistic paradigm, the aim of which is to understand and further investigate the phenomena, starting with education as a social construct that is subject to subjective interpretations and to the meanings given to it by the researchers involved in the study [53]. However, from the perspective that is proposed, this work could tend towards a socio-critical paradigm since the ultimate aim of this type of research is the improvement of initial teacher training, taking the teacher as a key part of the teaching and learning processes.

The overall aim of this work and its corresponding specific objectives are as follows:

- General aim (GA): To identify pre-service teachers' conceptions of gamification as a teaching methodology to address controversial heritages.

  Specific objectives (SO):

- SO 1. Investigate the conceptions of student teachers in initial teacher training on how to approach RSPs based on controversial heritages.
- SO 2. Detect the methodological strategies known and valued by pre-service teachers for heritage education.
- SO 3. Determine the degree of the initial teacher-training student teachers' mastery of the gamified methodological strategies.
- SO 4. Establish correlations between the knowledge of controversial heritage, the educational use of RSPs, and the intention to use gamified methodological strategies in modifying the conceptions of student teachers in initial training.

Therefore, the research questions, which are intended to be answered along with the achievement of the proposed objectives, are the following:

- Research Question (RQ) 1. How do pre-service teachers conceive the treatment of RSPs, based on controversial heritage?
- RQ 2. What are the most appropriate methodological strategies for teaching about heritage and controversy?
- RQ 3. What is the level of knowledge of pre-service teachers about gamification?
- RQ 4. How do knowledge regarding controversial heritage, the educational use of RSPs, and gamified methodological strategies come together?

### 2.1. Participants

The study involved 235 students—60 men (25.5%) and 175 women (74.5%) (Figure 1a)—studying for a degree in primary education at the University of Huelva, the University of Seville, and the University of Santiago de Compostela, selected as a non-probabilistic convenience sample [54] and chosen on the basis of the participation of these universities in the research project within which this work is framed.

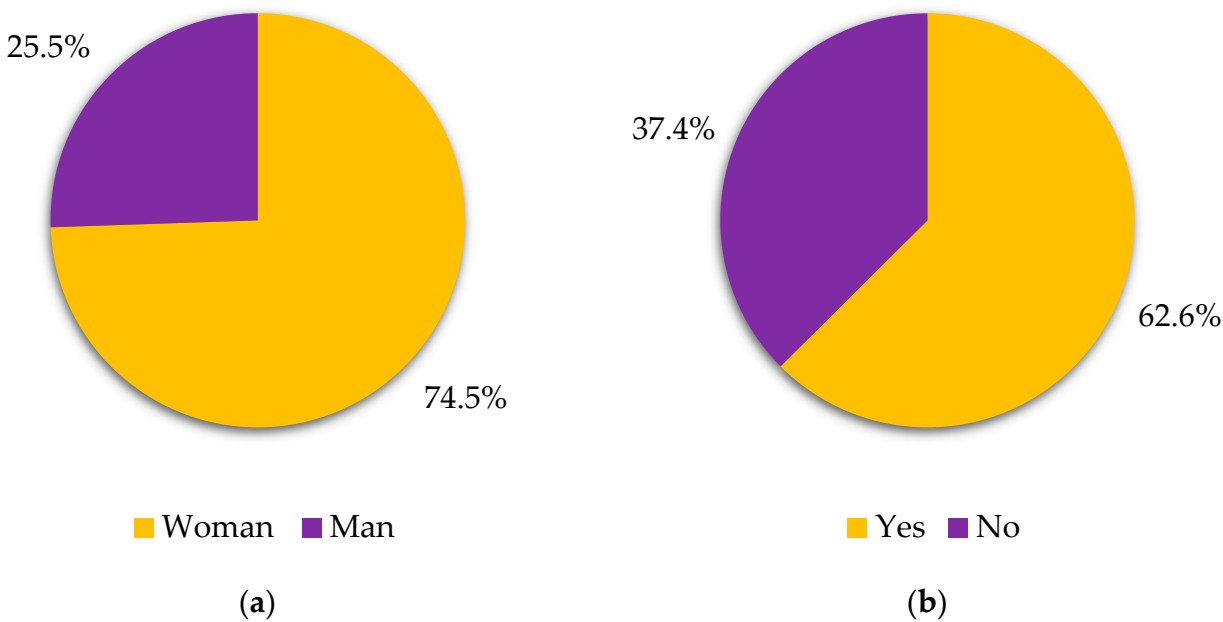

(**a**)  (**b**)

**Figure 1.** Participant characteristics: (**a**) gender distribution; (**b**) HE training.

Of the total sample, 147 initial teacher trainees, or 62.6%, stated that they had received HE training at different levels of education (Figure 1b); this may determine, to some extent, the subsequent results.

*2.2. Instruments*

To achieve the proposed objectives, the research design is characterized by the use of quantitative research techniques, such as the use of a questionnaire and correlational analysis using SPSS version 27 software, along with an analysis and subsequent interpretation, based on a system of categories adapted from other studies (Table 1) in which the indicators of the subcategories have different levels of definition that act as hypotheses regarding progression from levels of formulation ranging from the simple to the complex.

**Table 1.** Category system: information analysis tool.

| Categories | Subcategories | Indicators |
|---|---|---|
| I. Controversial heritages and RSPs | 1. Controversial themes | 1.1. Scope |
| | | Superficial |
| | | Inherent |
| | | 1.2. Temporality |
| | | Rooted |
| | | Current |
| | | 1.3. Context |
| | | Local |
| | | Global |
| | 2. Contravention of heritageperspectives | Anti-heritage |
| | | Heritage of cruelty |
| | | Interested heritage |
| | | Heritage with a gender perspective |
| | | Inclusive heritage |
| | | Subjected—rescued heritage |
| | | Heritage in transition |

**Table 1.** *Cont.*

| Categories | Subcategories | Indicators |
| --- | --- | --- |
| II. Controversial heritage teaching | 3. Use of controversial heritage | No socio-educational interest<br>Anecdotal utilization<br>Teaching resources<br>Full integration |
| | 4. Resources and strategies applied | Masterclasses<br>Specific materials<br>Chats and discussions<br>Visits<br>Experiments and simulations<br>Gamification |
| III. Gamification in the teaching of controversial heritage | 5. Gamified strategy types | Static role-playing games<br>Videogames<br>Breakouts<br>Escape rooms<br>Historical re-enactments<br>Live role-playing games<br>Civil dialogs |
| | 6. Use of the strategy for teaching controversial heritage | No educational interest<br>Lack of knowledge of its use<br>Aseptic didactic resource<br>Innovative strategy |

Source: Own creation, based on EPITEC2 [3], from works by Kerr and Huddleston [18], Arroyo et al. [7], Sampedro-Martín and Estepa [42], and Sampedro-Martín et al. [20].

The information-gathering instrument is a questionnaire drawn up ad hoc, consisting of 2 initial identification questions (gender and university), 3 dichotomous questions, associated with HE training and the respondent's familiarity with gamification as a teaching and learning strategy, 1 question asking the respondent to select images referring to heritage elements that may generate controversy (question 3), and 8 questions that have answers on a Likert scale from 1 to 5. Of these 8 questions, questions 1, 2, and 4 ascertain the respondent's knowledge and appreciation of controversial topics and controversial assets, while questions 5 and 6 tackle the use of controversial assets in the primary education classroom and the educational resources preferred for this purpose, and the last 3 questions examine the degree of knowledge and appreciation of gamified strategies on the part of the student teachers in initial teacher training. At this point, it should be noted that question 9 will need special treatment, since the option "I don't know it" has been included in the Likert scale from 1 to 5 to distinguish whether the respondents' answers are consistent with the knowledge of the gamified activities presented in it, and that they are not merely giving them a lower score in order of importance due to their lack of knowledge of them.

This questionnaire was first validated by four teachers/researchers, who are experts in HE and members of the EPITEC2 research team, and, subsequently, by means of a pilot test completed by 7 students—excluded from the final sample—who were taking a degree in primary education at the University of Huelva. The recommendations and proposed modifications from both validation stages were applied to the survey until the final questionnaire, which is presented in this paper, was produced (Table 2).

**Table 2.** The questionnaire's question statements and their associated categories.

| Nº | Questions | Cat. | Subcat. |
|---|---|---|---|
| - | Sex: | - | - |
| - | University: | - | - |
| - | Have you received training in heritage education? | - | - |
| - | In your role as a pupil, were you taught using gamification? | - | - |
| - | As a teacher, would you plan work units using gamification? | - | - |
| 1 | Please rate the following statements from 1 to 5, where 1 is "Totally disagree" and 5 is "Totally agree": <br> - Heritage can cause controversy, conflict, or debate. <br> - It is useful to address issues in the classroom that generate debate or controversy. <br> - Direct contact with heritage elements is the only way for learning to take place. <br> - Heritage and controversy are unrelated concepts. <br> - It is important that students learn to analyze heritage and their own culture from a gender perspective. <br> - The use of active methodological strategies is the only way to deal with classroom diversity. <br> - The defense of heritage generates exclusionary nationalism. <br> - Working with conflict in the classroom causes problems (with families, students, the school's ideology, etc.). | I | 1, 2 |
| 2 | In your opinion, how important do you think the following issues are in the current society? Please rate from 1 to 5, where 1 is "Not very important" and 5 is "Very important": <br> - Gender-based violence <br> - Racism <br> - LGTBQI phobia <br> - War <br> - Terrorism <br> - Colonization <br> - Climate change <br> - Deforestation <br> - Pollution <br> - Animal abuse <br> - Inequalities. | I | 1 |
| 3 | Mark the heritage elements you consider controversial: <br> - Encarnación Square (Seville, Spain) <br> - Spanish Civil War shelters (Almeria, Spain) <br> - Zara shop in the ancient convent of San Antonio el Real (Salamanca, Spain) <br> - National Lynching Memorial (Alabama, USA) <br> - Mapuche goldsmiths (Chile) <br> - Monument to Cristopher Columbus (Huelva, Spain) <br> - The Sao Domingos mine (Bejo, Portugal) <br> - Semana Santa <br> - *Henry Ford Hospital*, painted by Frida Kahlo <br> - Mauthausen concentration camp (Austria) <br> - LGBT Pride Day <br> - Padaung (the 'giraffe women' from Birmania) <br> - Bullfighting. | I | 1, 2 |

**Table 2.** *Cont.*

| Nº | Questions | Cat. | Subcat. |
|---|---|---|---|
| 4 | Please rate from 1 to 5 how controversial these groups of items are, where 1 is "Not controversial" and 5 is "Very controversial": <br><br> - Places or monuments that represent counter-values or that are the result of human atrocities, wars, persecution, and repression <br> - Traditions that involve some form of violence against humans or animals <br> - Protected natural areas threatened by human action <br> - Heritage elements that involve a conflict of political, ideological, economic, and/or social interests <br> - Heritage symbols or manifestations represented by social movements claiming equality/equity for discriminated, oppressed, or marginalized groups <br> - Elements that highlight the value of heritage objects made by women or that extol gender equality values <br> - Manifestations or elements claiming the need for access to and representation of diversity <br> - Heritage elements that have been modified for economic benefit <br> - Symbols or elements of a dominant culture that have displaced or subjugated communities, cultures, or social groups <br> - Heritage that was made invisible or destroyed and that is now being enhanced <br> - Heritage manifestations that exemplify, represent, or reinforce gender roles and stereotypes or instrumentalize, sexualize, and dehumanize women's bodies <br> - Forgotten or abandoned elements that have been given new value by adapting them to the social demands of today. | I | 2 |
| 5 | How often would you use heritage elements in the classroom? | II | 3 |
| 6 | Which of the following activities seem most interesting for tackling controversial issues when teaching about heritage? (Rate each of them from 1 to 5, where 1 is "Not at all interesting" and 5 is "Very interesting") | II, III | 4, 5, 6 |
| 7 | Indicate your degree of knowledge of gamification: | III | 5, 6 |
| 8 | Indicate how useful you consider gamification for the classroom teaching process: | III | 5, 6 |
| 9 | Which of the following activities do you consider most suitable for working on controversial issues when teaching about heritage? Please rate from 1 to 5, where 1 is "Not suitable" and 5 is "Very suitable": | I, II, III | 1, 2, 3, 4, 5, 6 |

Source: Own creation.

A total of 44 variables were analyzed, which were included in the questions with Likert scale answers from 5 to 1, to which were added the 3 variables of the respective dichotomous questions with which the hypothesis tests were carried out. These 44 variables have been classified in relation to the system of categories, to ensure the correct processing of the information and a clearer presentation of the results (Table 3).

**Table 3.** The relationship between questions, variables, categories, and their associated subcategories.

| Q nº | Variables | Cat. | Subcat. |
|---|---|---|---|
| 1 | ControHeri, ClassDebate, DireConta, Disengagement, StuGender, MethoDiversity, Nationalism, ClassConflict | I | 1,2 |
| 2 | GenderVio, Racism, LGTBI, War, Terrorism, Antidemocratic, Colonisation, ClimateChange, Deforestation, Contamination, AnimalVio, Inequality | I | 1 |
| 4 | Anti-heritage, Cruelty, InterestedA, InterestedB, InclusiveA, GenderA, InclusiveB, TransitionA, Subjected, TransitionB, GenderB, TransitionC | I | 2 |
| 5 | Frequency | II | 3 |
| 6 | Book, Debate, Visit, Gamification, Explanation, Social Networks, Slide, Re-enactments, Video | II, III | 4, 5, 6 |
| 7 | KnowGami | III | 5, 6 |
| 8 | UseGami | III | 5, 6 |

Source: Own creation. Note: There are not any quantitative variables in questions 3 and 9.



Likewise, the reliability tests of the SPSS version 27 tool were applied to the questionnaire, obtaining a satisfactory internal consistency, as determined by Cronbach's α, of 0.859. On the other hand, the McDonald's Ω test could not be performed because some of the questions in the questionnaire are consciously and intentionally formulated in reverse, in order to ascertain the consistency of the prospective teachers' answers.

Regarding the tests on the sample, the sample distribution was found to be non-normal since, when applying the Kolmogorov–Smirnov test, the null hypothesis was rejected (<0.001), i.e., all distributions are non-normal. Similarly, once the factor analysis was carried out using the KMO test, the result indicated that the sample adequacy was convenient, reaching a value of 0.806.

## 3. Results

In terms of the structure of the questionnaire, firstly, non-parametric tests were applied to relate the three initial dichotomous questions—those related to training in HE, experience as a student with gamified methodological strategies, and the respondent's intention to teach through gamification in their professional future as a primary education teacher—with the rest of the variables of the instrument.

The first hypothesis test, which was carried out using the Mann–Whitney U test, the purpose of which is to compare sample means for each item, confronted the initial HE training with all the variables of the questionnaire, finding a high degree of significance with three variables from the questionnaire (Table 4). Firstly, a significance level of 0.027 was obtained with the variable 'ControHeri', an effect size of 0.09 (>0.05) and a statistical power of 11% (<20%), which implies a significant difference between the responses of those students who received HE training during their academic life (M = 4.14) and those who did not (M = 3.89). In this case, students who had experienced HE teaching and learning processes stated that heritage can give rise to controversy, conflict, or debate, indicating that they have some awareness that RSPs can underlie the heritage manifestations of communities.

**Table 4.** Non-parametric tests of the hypothesis variable, 'HE training'.

|  | Mann–Whitney U | Asymptot. Sig. (Bilateral) | Statistical Power | Effect Size | Mean "No" (0) *n* = 88 | Standard Dev. (0) *n* = 88 | Mean "Yes" (1) *n* = 147 | Standard Dev. (1) *n* = 147 |
|---|---|---|---|---|---|---|---|---|
| ControHeri | 5414.500 | 0.027 | 11% | 0.09 | 3.89 | 0.964 | 4.14 | 0.972 |
| Visit | 5375.000 | 0.002 | 40% | 0.45 | 4.64 | 0.610 | 4.85 | 0.411 |

Source: Own creation.

Furthermore, the degree of significance of this item with the 'Visit' variable is 0.002, which indicates that despite obtaining a statistical power of 35% (>20%) and an effect size of 0.52 (>0.05), there is a notable difference between the responses of HE-trained students (M = 4.85) and those who were not previously HE-trained (M = 4.64). This difference implies that HE-trained students value making visits to heritage contexts and direct contact with heritage elements more positively than those without HE training.

The second Mann–Whitney U test that was performed sought to compare the sample means and standard deviations for all variables regarding the dichotomous question that referred to the initial teacher trainee's experience with gamification as a student throughout their academic career. Significant differences were found between having been taught or not taught through gamified strategies in the case of two variables of the questionnaire (Table 5), but the statistical power of both differences is higher than 20%, which seems to indicate that this difference is not relevant for this study.

**Table 5.** The non-parametric variable 'StuGami' hypothesis tests.

| | Mann-Whitney U | Asymptot. Sig. (Bilateral) | Statistical Power | Effect Size | Mean "Yes" (1) n = 65 | Standard Dev. (1) n = 65 | Mean "No" (2) n = 170 | Standard Dev. (2) n = 170 |
|---|---|---|---|---|---|---|---|---|
| Anti-heritage | 4262.000 | 0.004 | 35% | 0.37 | 3.72 | 0.944 | 4.08 | 1.023 |
| TransitionA | 4738.000 | 0.077 | 45% | 0.24 | 3.68 | 1.077 | 3.94 | 1.086 |

Source: Own creation.

The following non-parametric tests were carried out with respect to the dichotomous variable, 'TeachGami', to detect significant differences between the responses of those teachers in initial training who showed their predisposition to implement the teaching and learning proposals with gamified methodological strategies in their professional future ($n = 231$) and those who stated their refusal to teach through gamification ($n = 4$). Although, when calculating the statistical power and effect size, the results showed that the differences between the variables were not significant, some differences were found in the Mann–Whitney U test that we consider to be noteworthy for interpreting the results of this study (Table 6).

**Table 6.** The non-parametric variable 'TeachGami' hypothesis tests.

| | Mann-Whitney U | Asymptot. Sig. (Bilateral) | Statistical Power | Effect Size | Mean "Yes" (1) n = 231 | Standard Dev. (1) n = 231 | Mean "No" (2) n = 4 | Standard Dev. (2) n = 4 |
|---|---|---|---|---|---|---|---|---|
| Gamification | 148.500 | 0.006 | 62% | 1.55 | 4.58 | 0.568 | 3.75 | 0.500 |
| Re-enactments | 233.000 | 0.056 | 54% | 1.02 | 4.44 | 0.707 | 3.25 | 1.500 |
| KnowGami | 91.000 | 0.003 | 75% | 1.86 | 3.50 | 0.796 | 2.00 | 0.816 |
| UseGami | 86.000 | 0.002 | 76% | 1.93 | 4.45 | 0.677 | 3.00 | 0.816 |

Source: Own creation.

Comparing the responses of all items with the dichotomous variable 'TeachGami', on the one hand, it was found that students who said that they would use gamification in their future work as teachers considered that gamified activities (M = 4.58) and historical re-enactments (M = 4.44) were very interesting for the treatment of controversial topics in heritage teaching, compared to those who stated their refusal to teach through gamified strategies; they valued gamification (M = 3.75) and re-enactments (M = 3.25) more negatively in the context of HE work through RSPs.

On the other hand, there were also notable differences regarding the variable 'TeachGami', in terms of the degree of knowledge that the respondents claimed to possess about gamification ((1), M = 3.50; (2), M = 2.00) and, consequently, in the usefulness that they attributed to gamified strategies for the teaching and learning processes of controversial heritage ((1), M = 4.45; (2), M = 3.00).

Once the non-parametric tests had been carried out, an analysis was made of the bivariate correlations between the 44 variables in the questionnaire and the categories presented in Table 1, finding 291 correlations that were significant at 0.01 and 121 at 0.05. Although correlations with values lower than 0.05 are valid in the social sciences, only those with values below 0.01 were selected for this study, as they have an $\alpha$-error of less than 1%, i.e., the degree of significance of these correlations is 99%.

*3.1. Category I: Controversial Heritages and RSPs*

Regarding the first category of analysis, relating to controversial issues and controversial heritage, there are variables with numerous significant bivariate correlations from Spearman's Rho test that are of interest for this study.

The 'ControHeri' variable (Table 7) of subcategory I.1., controversial issues, presents a correlation coefficient of 0.475 and a bilateral significance—and an $\alpha$-error of less than

0.001—with the variable 'ClassDebate', which implies that those who agreed that heritage can generate controversy, conflict, or debate also stated that it is convenient to introduce issues in the classroom that can generate conflict or debate.

**Table 7.** Bivariate correlations with the 'ControHeri' variable (subcategory I.1).

| Spearman's Rho | | | |
|---|---|---|---|
| **Variables** | **Value** | **Sig (Bilateral)** | ***n* (Valid Case)** |
| ClassDebate | 0.475 | 0.000 | 235 |
| Disengagement | −0.337 | 0.000 | 235 |
| StuGender | 0.184 | 0.005 | 235 |
| LGTBI | 0.193 | 0.003 | 235 |
| Anti-heritage | 0.301 | 0.000 | 235 |
| Cruelty | 0.220 | 0.000 | 235 |
| InterestedA | 0.201 | 0.002 | 235 |
| InterestedB | 0.366 | 0.000 | 235 |
| TransitionA | 0.235 | 0.000 | 235 |
| Subjected | 0.317 | 0.000 | 235 |
| GenderB | 0.263 | 0.000 | 235 |
| TransitionC | 0.203 | 0.002 | 235 |
| Re-enactments | 0.186 | 0.004 | 235 |
| UseGami | 0.273 | 0.000 | 235 |

Source: Own creation.

Likewise, 'ControHeri' has a correlation coefficient of −0.0337 and a bilateral significance degree—along with an $\alpha$-error of less than 0.001—with the 'Disengagement' variable, meaning that those who gave a high score to the first variable gave a low score to the second variable, which item expressed the view that heritage and controversy are totally unrelated concepts.

On the other hand, this variable, 'ControHeri', correlates with the variables of 'Re-enactments' (a correlation coefficient of 0.186 and a degree of significance with an $\alpha$ error of 0.004) and 'UseGami' (correlation coefficient of 0.273 and a degree of significance with an $\alpha$-error of less than 0.001), which indicates that those pre-service teachers who are aware that heritage can generate some kind of controversy or dilemma do positively value the use of gamification as a teaching and learning strategy and consider historical re-enactments to be suitable activities for dealing with controversial heritage in a primary school classroom.

With regard to subcategory I.2., controversial heritage perspectives, the variable with the highest number of bivariate correlations is 'InterestedB' (Table 8), a variable that corresponds to the item referring to the degree of controversy that may be caused by those heritage elements that involve a conflict of political, ideological, economic, and/or social interests.

For the purposes of this study, the correlations established between 'InterestedB' and 'ClassDebate' (a correlation coefficient of 0.324 and a degree of bilateral significance with an $\alpha$-error of less than 0.001), 'InterestedB' and 'Re-enactments' (a correlation coefficient of 0.324 and a degree of bilateral significance with an $\alpha$-error of less than 0.001), and 'InterestedB' and 'UseGami' (a correlation coefficient of 0.190 and a degree of bilateral significance with an $\alpha$-error of 0.003) demonstrate the respondents' perspectives regarding controversial heritage and the methodological strategies that pre-service teachers deem suitable in the HE context.

It should be noted that the initial teacher trainees selected those heritage elements that they considered most controversial in question 3 of the questionnaire. The results (Figure 2) reveal that bullfighting is seen as the most controversial element of all (95.7%).

**Table 8.** Bivariate correlations of the 'InterestedB' variable (subcategory I.2).

| | Spearman Rho | | |
|---|---|---|---|
| **Variables** | **Value** | **Sig (Bilateral)** | ***n* (Valid Case)** |
| ControHeri | 0.366 | 0.000 | 235 |
| ClassDebate | 0.324 | 0.000 | 235 |
| Disengagement | −0.215 | 0.000 | 235 |
| StuGender | 0.178 | 0.006 | 235 |
| GenderVio | 0.309 | 0.000 | 235 |
| Racism | 0.285 | 0.000 | 235 |
| LGTBI | 0.261 | 0.000 | 235 |
| War | 0.290 | 0.000 | 235 |
| Terrorism | 0.225 | 0.000 | 235 |
| Antidemocratic | 0.233 | 0.000 | 235 |
| ClimateChange | 0.293 | 0.000 | 235 |
| Deforestation | 0.210 | 0.001 | 235 |
| Contamination | 0.259 | 0.000 | 235 |
| AnimalVio | 0.231 | 0.000 | 235 |
| Inequality | 0.264 | 0.000 | 235 |
| Anti-heritage | 0.370 | 0.000 | 235 |
| Cruelty | 0.373 | 0.000 | 235 |
| InterestedA | 0.328 | 0.000 | 235 |
| InclusiveA | 0.397 | 0.000 | 235 |
| GenderA | 0.308 | 0.000 | 235 |
| InclusiveB | 0.254 | 0.000 | 235 |
| TransitionA | 0.324 | 0.000 | 235 |
| Subjected | 0.388 | 0.000 | 235 |
| TransitionB | 0.247 | 0.000 | 235 |
| GenderB | 0.399 | 0.000 | 235 |
| TransitionC | 0.197 | 0.002 | 235 |
| Re-enactments | 0.186 | 0.004 | 235 |
| UseGami | 0.190 | 0.003 | 235 |

Source: Own creation.

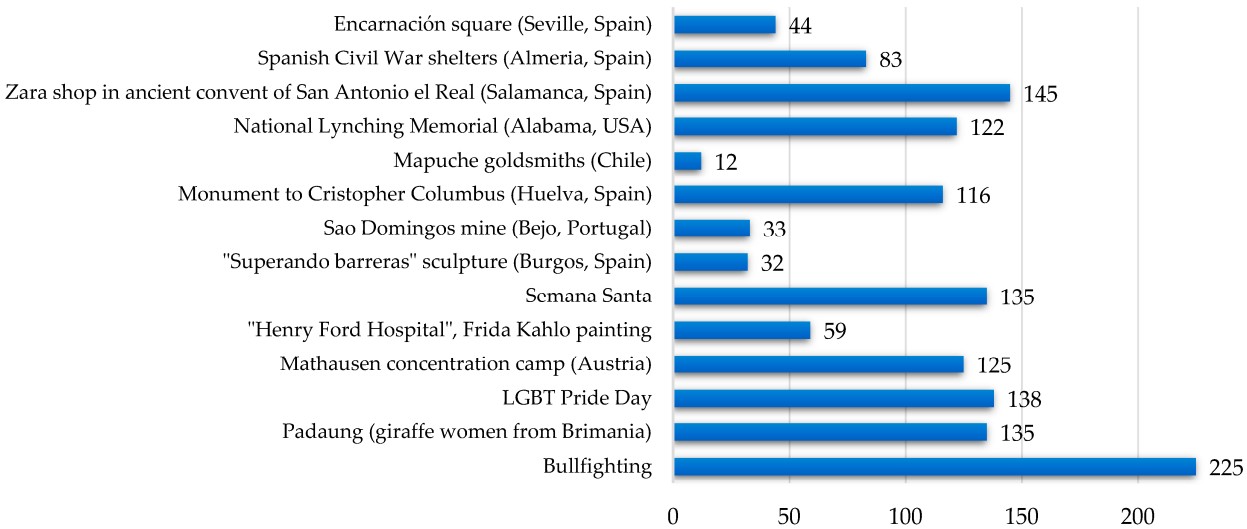

**Figure 2.** Answers to the question on examples of controversial heritage assets (subcategory I.2).

*3.2. Category II: Teaching Controversial Heritage*

From the second analysis category of this work, which refers to the possibilities of using controversial heritage in the classroom and the resources and methodological strategies used for its treatment, some correlations that are of great interest for this study are those related to the 'Frequency' variable, which corresponds to the item that questions

the frequency with which student teachers in initial training would use heritage elements in the primary school classroom (Figure 3).

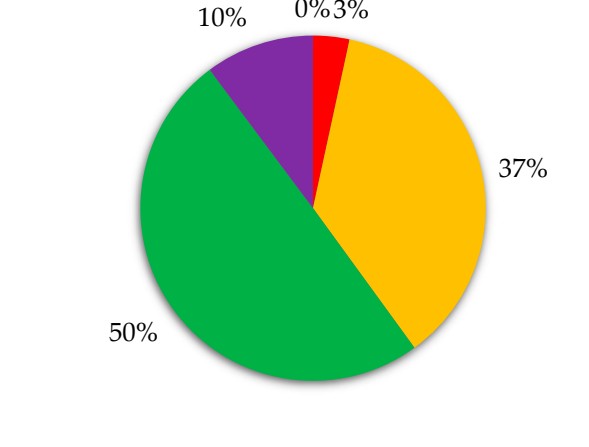

**Figure 3.** The respondents' predisposition regarding the frequency of the use of heritage in the classroom.

The 'Frequency' variable correlates significantly with the variables (Table 9) in subcategory I.1, controversial issues, 'GenderVio' (a correlation coefficient of 0.194 and a degree of bilateral significance with an $\alpha$-error of 0.003), 'Racism' (a correlation coefficient of 0.174 and a degree of bilateral significance with an $\alpha$-error of 0.008), and 'Colonization' (a correlation coefficient of 0.182 and a degree of bilateral significance with an $\alpha$-error of 0.005). Similarly, this variable correlates with the variables relating to perspectives on controversial heritages, subcategory I.2., 'GenderB' (a correlation coefficient of 0.218 and a degree of bilateral significance with an $\alpha$-error of less than 0.001), and 'TransitionC' (a correlation coefficient of 0.218 and a degree of bilateral significance with an $\alpha$-error of 0.001). These correlations could indicate that student teachers in initial teacher training who consider gender-based violence, racism, and the occupation of territories and colonization to be relevant issues in our society intend to use heritage elements in the classroom quite frequently. Similarly, those who attribute some controversy to heritage manifestations that instrumentalize women's bodies and to heritage elements that have been enhanced after a period of neglect also express their intention to make frequent use of heritage in primary education.

**Table 9.** Bivariate correlations of the 'Frequency' variable (subcategory II.3).

| | Spearman Rho | | |
|---|---|---|---|
| **Variables** | **Value** | **Sig (Bilateral)** | **$n$ (Valid Case)** |
| GenderVio | 0.194 | 0.003 | 235 |
| Racism | 0.174 | 0.008 | 235 |
| Colonization | 0.182 | 0.005 | 235 |
| GenderB | 0.218 | 0.000 | 235 |
| TransitionC | 0.212 | 0.001 | 235 |
| Visit | 0.178 | 0.006 | 235 |
| Gamification | 0.184 | 0.005 | 235 |
| Re-enactments | 0.181 | 0.005 | 235 |
| Video | 0.188 | 0.004 | 235 |

Source: Own creation.

Likewise, the 'Frequency' variable correlates significantly with the variables of 'Gamification' (a correlation coefficient of 0.181 and a degree of bilateral significance with an $\alpha$-error of 0.005) and 'Re-enactments' (a correlation coefficient of 0.188 and a degree of

bilateral significance with an $\alpha$-error of 0.004), both from subcategory II.4., referring to the resources and strategies used in HE. These correlations indicate that those future teachers who show a predisposition to use heritage as a teaching object in their proposals often give positive ratings to historical re-enactments and gamified activities when dealing with controversial topics associated with heritage.

In turn, the 'Gamification' variable, from subcategory II.4., correlates with a high degree of significance with variables of the same category, as well as with the 'Subjected' variable from subcategory I.2., the correlation coefficient of which is 0.171 and the degree of bilateral significance has an $\alpha$-error value of 0.009. In this case, the high score for the use of gamified activities correlates significantly with the consideration that heritage elements display the symbols of a dominant culture that has displaced or subjugated communities, cultures, or social groups.

Similarly, the correlations of the 'Gamification' variable (Table 10) indicate that prospective teachers who value gamification positively for the purposes of teaching controversial heritage also have high regard for visits to heritage contexts, historical re-enactments, and the educational use of social networks for the same purpose. In addition, a correlation is established with the 'Book' variable (a correlation coefficient of $-0.195$ and a degree of bilateral significance with an $\alpha$-error of 0.003) as those students who gave high scores to the gamified activities gave very low scores to the textbook as a resource to tackle controversial topics associated with heritage.

**Table 10.** Bivariate correlations of the 'Gamification' variable (subcategory II.4.).

| | Spearman Rho | | |
|---|---|---|---|
| **Variables** | **Value** | **Sig (Bilateral)** | ***n* (Valid Case)** |
| Submitted | 0.171 | 0.009 | 235 |
| Frequency | 0.184 | 0.005 | 235 |
| Book | $-0.195$ | 0.003 | 235 |
| Visit | 0.244 | 0.000 | 235 |
| Social Networks | 0.225 | 0.000 | 235 |
| Re-enactments | 0.184 | 0.005 | 235 |
| KnowGami | 0.328 | 0.000 | 235 |
| UseGami | 0.519 | 0.000 | 235 |

Source: Own creation.

### 3.3. Category III: Gamification in Teaching Controversial Heritage

In the third category of analysis in this study, which relates to knowledge and perspectives on the use of gamification for HE work based on controversial heritage, the results of subcategory III.5, representing the type of gamified strategies that initial teacher trainees consider educational techniques with high potential, include live role-playing games (65.1%), historical re-enactments (61.7%), gymkhanas (56.2%), debate simulations (52.7%) and escape rooms (52.3%), giving them a score of 5 on the Likert scale (Figures 4–6). In contrast, 66.4% of participants claimed to be unaware of the Breakout tool.

In subcategory III.6. regarding the usefulness of gamification for teaching controversial heritage, we found that the 'UseGami' variable correlates with a high degree of significance with items in the same category, and correlates with 'KnowGami' with a correlation coefficient of 0.444 and a bilateral degree of significance with an $\alpha$ error of less than 0.001, as well as with variables from categories I and II (Table 11). The correlation established between 'KnowGami' and 'UseGami' indicates that future teachers who claim to have some in-depth knowledge of gamified educational strategies consider these gamification activities to be very useful in the teaching and learning processes in primary school classrooms.

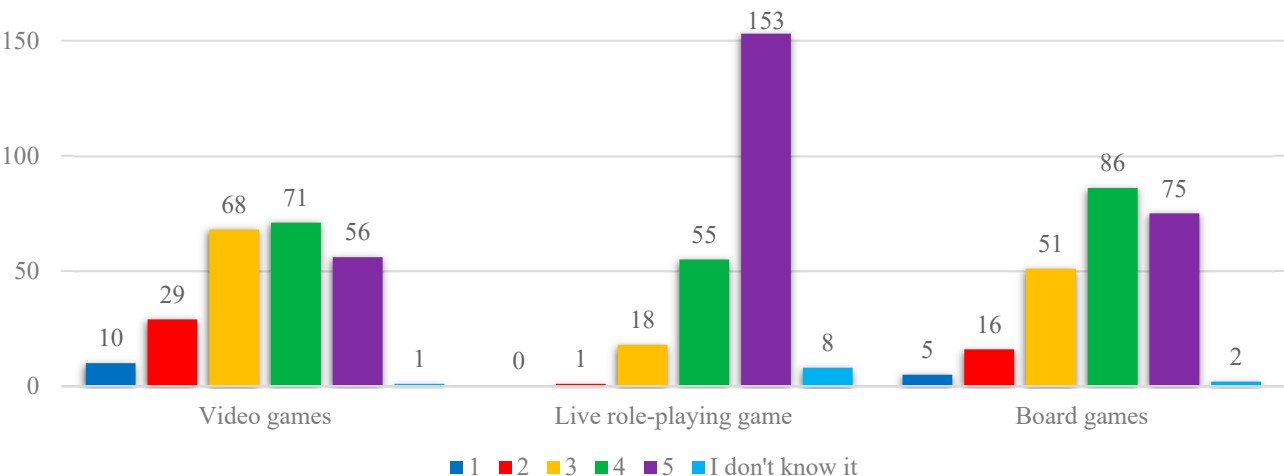

**Figure 4.** Video game, live role-playing game, and board game ratings (subcategory III.5).

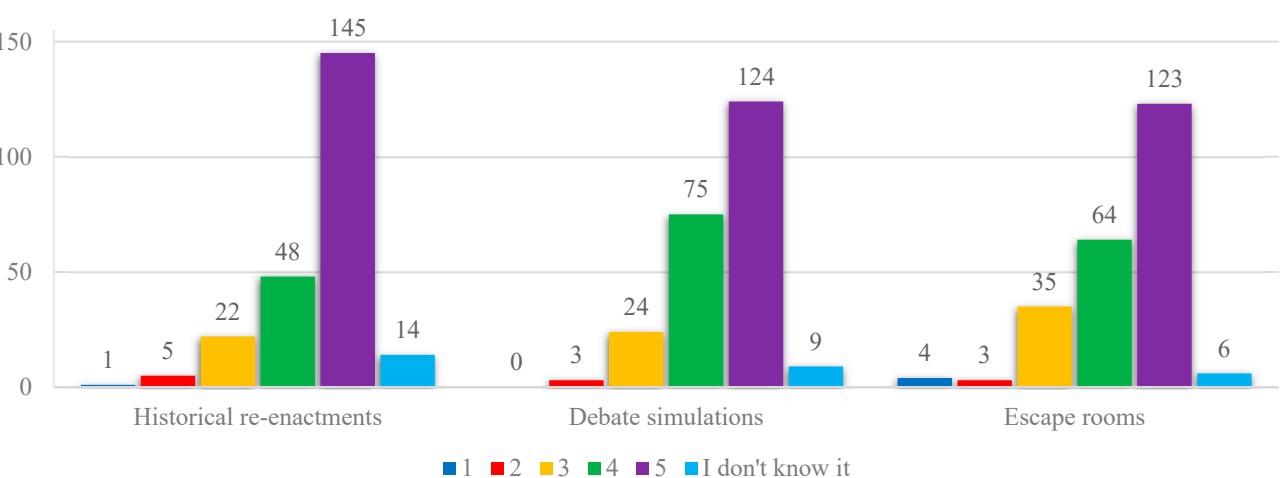

**Figure 5.** Historical re-enactment, debate simulation, and Escape Room ratings (subcategory III.5).

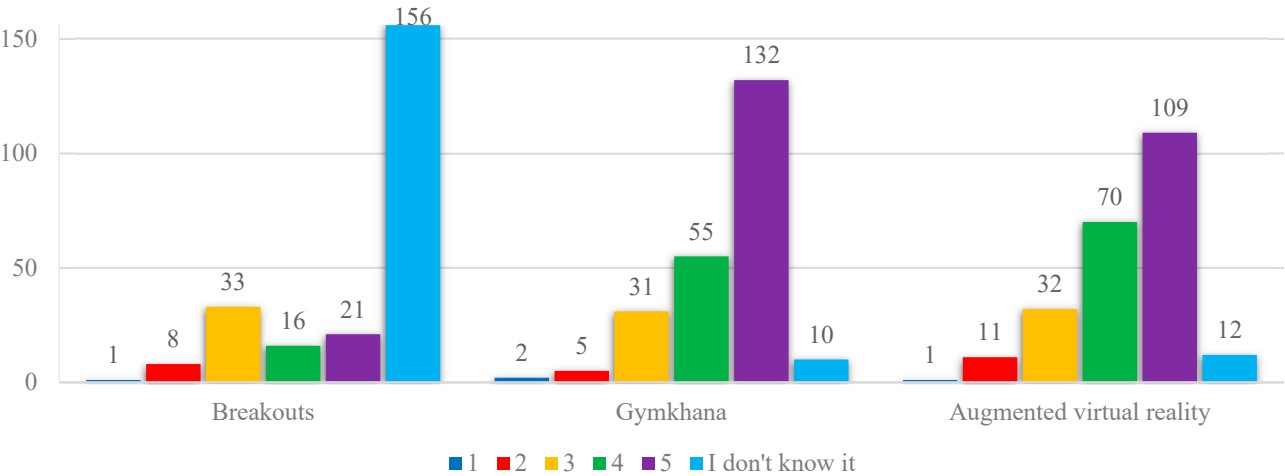

**Figure 6.** Breakout, gymkhana, and augmented virtual reality ratings (subcategory III.5).

**Table 11.** Bivariate correlations of the 'UseGami' variable (subcategory III.6).

| | Spearman Rho | | |
| --- | --- | --- | --- |
| **Variables** | **Value** | **Sig (Bilateral)** | ***n* (Valid Case)** |
| ControHeri | 0.273 | 0.000 | 235 |
| ClassDebate | 0.260 | 0.000 | 235 |
| Disengagement | −0.272 | 0.000 | 235 |
| Cruelty | 0.186 | 0.004 | 235 |
| InterestedB | 0.190 | 0.003 | 235 |
| GenderB | 0.211 | 0.001 | 235 |
| Book | −0.233 | 0.000 | 235 |
| Visit | 0.173 | 0.008 | 235 |
| Gamification | 0.519 | 0.000 | 235 |
| Re-enactments | 0.192 | 0.003 | 235 |
| KnowGami | 0.446 | 0.000 | 235 |

Source: Own creation.

The correlations presented herein show that pre-service teachers who consider gamification an educational strategy with great educational potential and utility are aware that heritage can provoke controversy and conflict ('UseGami' correlates with 'Disengagement' with a correlation coefficient of −0.272 and a bilateral significance with an α-error of less than 0.001). They say that it is advisable to introduce issues that will generate debate in the primary classroom ('UseGami' correlates with 'ClassDebate' with a correlation coefficient of 0.260 and a bilateral significance with an α-error of less than 0.001).

Conversely, the 'UseGami' variable correlates with a high degree of significance to three variables in subcategory I.2., which refers to controversial heritage perspectives, demonstrating the high level of consideration given to the usefulness of gamification in the case of festivities or traditions that involve some kind of violence against humans or animals ('Cruelty' variable), heritage elements that entail a conflict of political, ideological, economic, and/or social interests ('InterestedB' variable), and heritage manifestations that perpetuate gender roles and stereotypes or that instrumentalize women's bodies ('GenderB' variable), viewing these as heritage elements that generate a high degree of controversy ('GenderB' variable).

Finally, those participants who considered gamified educational strategies to be very useful, as indicated by the correlations, highly rated visits to heritage contexts, gamification activities, and historical re-enactments for the treatment of the controversial topics that underlie heritage. Conversely, these student teachers in initial teacher education gave a negative assessment of the use of the textbook for this purpose.

## 4. Discussion of Study Results and Conclusions

As this is an exploratory study, there are no antecedent studies that relate the concepts as they are established in this research. However, several research studies have been selected for their similarity in the treatment of some of the categories and subcategories of this study on teaching conceptions, in order to contrast the information obtained and, thus, support the relevance and pertinence of this research for future proposals. Therefore, the analysis will take into consideration other works carried out within the framework of the conceptions of student teachers in initial training: on heritage [27], controversial topics, and RSPs [18]; on the usefulness of their teaching [1] and the resources used [31]; on gamification or gamified tools [34] for teaching those topics related to this study, for example, using role-playing games to explore socio-scientific conflicts [41].

### 4.1. How Do Pre-Service Teachers Conceive the Treatment of RSPs Based on Controversial Heritage?

As noted in the previous section, the results show that pre-service teachers are aware that heritage can generate controversy, conflict, or debate—with 71.9% of responses ranging from 4 (agree) to 5 (strongly agree)—and know that it is appropriate to introduce such

debates in the primary classroom—with 87.7% of responses ranging from 4 (agree) to 5 (strongly agree). This contradicts the views of future secondary school teachers, who, according to the study by Ferreras et al. [1], understand heritage from an academic perspective that is far removed from the formation of an active citizenry.

However, the results of this study are in line with the findings of Cuenca et al. [27], who found in the work of future teachers that they addressed content related to RSPs and sustainability in their teaching proposals for HE and that they reflected on current societies based on controversial heritages.

This study has also shown that future teachers are concerned about various RSPs that can be associated with heritage events in their immediate environment. Thus, 80% attach great social importance to gender-based violence, 69% to racism, 72.8% to LGTBQI phobia, 66.8% to armed conflicts, 68% to climate change, 67.3% to pollution, and 68.5% to inequalities. These data could indicate that when taking into account their predisposition to introduce in the classroom those issues that give rise to debate or controversy, future teachers would carry out proposals on RSPs and some heritage manifestations that they consider controversial—such as bullfighting, selected by 95.7% of participants as a heritage element that generates controversy—at the primary education stage.

Conversely, in contrast with the results of studies by Estepa et al. [55] and Ferreras et al. [1], who reported that future primary and secondary teachers showed a lack of relevance regarding heritage in their teaching and learning proposals, 50% of the future primary teachers surveyed in this study declared their predisposition to use heritage 'almost always' in their classrooms, and 10% declared to do so 'always', just as was reported in the study by Sampedro-Martín [56], wherein more than 50% of the primary school teachers claimed to use heritage elements quite frequently.

### 4.2. What Are the Most Appropriate Methodological Strategies for Teaching Heritage and Controversy?

Although there are numerous studies that place the textbook as a priority resource for teaching heritage [27], in the specific case of the work presented in this report, the results show that 76.2% of future primary school teachers rated the textbook as an educational resource for teaching controversial topics based on heritage as either not at all interesting or not very interesting [27]. Moreover, 80.9% of the initial teacher trainees surveyed considered that visits to heritage contexts, museums, interpretation centers, archaeological sites, and natural spaces, among others, are activities of great interest for the teaching of the RSPs underlying heritage, compared to 38% of the sample studied by Ferreras et al. [1], who, despite mentioning activities in museums and virtual museums, see such events as a one-off practice not integrated into the curriculum.

On the one hand, the study by Ferreras et al. [1] showed that 47% of the future secondary teachers surveyed applied the didactic use of heritage in an enlightened approach, based on the transmission of theoretical knowledge through traditional activities wherein the role of the students is passive. On the other hand, regarding which resources and methodological strategies the future teachers in this study deem appropriate, the results show that student teachers in initial training consider a wide variety of activities and educational resources for teaching controversial heritage, valuing very positively those that promote active student participation. A clear example of this is the role-playing game, which, according to Zelaieta et al. [37], constitutes a novel and stimulating methodology for educational encounters, based on autonomous and investigative work situations. The participants in this study agreed with these authors that both live role-playing games (65.1%) and historical re-enactments (61.7%) or debate simulations (52.7%) (the latter two being variants of role-playing according to Sampedro-Martín and Estepa [42]) are positioned as the most appropriate activities for working on controversial issues in heritage teaching in the classroom. Along these lines, Kerr and Huddleston [18] propose as typical activities for teaching controversial topics those tasks that encourage the depersonalization of students, such as role-playing games, and promote debate.

In short, gamified activities that encourage debate, such as role-playing games, historical re-enactment, or debate simulation techniques such as Civil Dialogue [49], are tools designed to improve the training of future teachers; they encourage argumentation, position-taking, controversy, conflict, and debate, and are directly aligned with the aim of teaching students about controversial heritage [49].

*4.3. What Is the Level of Knowledge of Pre-Service Teachers about Gamification?*

In this study, the majority of pre-service teachers reported a medium-high level of knowledge of gamification—43.4% marked it as 4, and 37.4% responded with a 3. However, we can observe that in 66.4% of cases, they were unaware of one of the most useful gamified activities in the classroom, Breakout, as Moreno-Fuentes [34] states in his study, where he affirms that the acceptance of this tool by future teachers is high—28.2% rated it with a 4 and 53.8% gave it a 3. On the other hand, the same participants who claimed to have a certain degree of knowledge about gamification were those who denied (72.3%) having received any training on the subject or declared that they had not undergone, as students in the different educational stages, experiences in which their teachers used gamified activities in the teaching-learning process.

From these contradictions, it can be deduced that even though they claim to know about gamification, student teachers in initial training need more preparation in the use of tools of this type since, both in the study by Moreno-Fuentes [34] (87.2%) and in this one described here (98.3%), the students coincide in their intention to use gamified activities in their future teaching practice with primary pupils, hence showing the importance of giving them sufficient tools to plan work units in which they use gamification, as they wish to do.

Focusing on one of these gamified tools, the initial teacher trainees who took part in the study of Moreno-Fernández et al. [31] said, in 95% of the cases, that they know what an Escape Room consists of, which finding coincides with what the future teachers in this study stated, wherein they also considered the Escape Room to be one of the gamified activities with the greatest potential for working on controversial heritage issues in their classroom practice (52.3%).

*4.4. How Do the Knowledge of Controversial Heritage, the Educational Use of RSPs, and Gamified Methodological Strategies Come Together?*

This study shows that teachers who claim to have some in-depth knowledge of gamified educational strategies consider the use of these gamification activities and historical re-enactments in the teaching-learning processes in primary education to be very useful, especially for tackling controversial topics in heritage teaching.

According to Zelaieta et al. [37], "Students demand specific experiences to improve the learning of gamified activities [ . . . ] In this sense, and within experiential learning, the proposal of academic debates is considered ideal for initial teacher training" (p. 741). This is in line with Moreno-Fuentes [34], who notes that he finds it interesting in his research that 53.4% of student teachers said that they would need training or wished to find out more about this type of active methodology.

According to Cuenca et al. [27], the use of gamified activities such as video games or augmented reality, less valued in this study but equally taken into consideration—23.8% rated the video game with a 5 and 30.2% with a 4, while 46.4% rated augmented reality with a 5—among other proposals such as role playing or debates, which were more valued than the previous ones (88.5% and 84.7% rated them respectively between 4 and 5), demonstrates the potential of teachers as facilitators of learning and motivators of eco-social educational processes. The documents analyzed by these authors in their research show the importance of the curricular design of innovation proposals through a new selection of contents, such as controversial heritage, and interactive methodological aspects, such as gamified activities, among which the educational use of video games stands out. To this list, this study adds role-playing games, Escape Rooms, Breakouts, and Civil Dialog [48], providing a new way

of understanding interaction in the processes of teaching and learning about heritage for citizenship education and approaches to RSPs and sustainability.

From the results obtained, it can be seen how student teachers in initial training consider the introduction of gamification in the classroom to be very useful for working on heritage elements that involve a conflict of environmental, political, ideological, economic, and/or social interests, which are known as interested heritage [7,20]. However, by showing concrete examples of such heritage elements of interest, such as the mushrooms in the Plaza de la Encarnación in Seville (Spain) or the Minas de Sao Domingos in Bejo (Portugal), future teachers fail to appreciate the controversy behind these heritage manifestations. This result could denote a lack of reflection and a critical view of specific heritage elements, which reinforces the idea that there is a need for HE training designed to foster critical thinking. This is evident in the work of Cuenca et al. [27], where it can be seen that the productions of future HE-trained teachers and their research were characterized by more evolved and complex activity designs, along with a higher degree of theoretical-practical reflection and the inclusion of gamified methodological strategies.

For this reason, it is essential to introduce in initial teacher training an in-depth study of the active methodological aspects that manage to connect the functional nature of teaching with the motivational nature of learning since, as Lorca et al. [41] noted, teacher trainers should incorporate in their classroom designs specific strategies to overcome the epistemological, affective, and contextual obstacles that commonly prevail in initial teacher training, such as poor knowledge of scientific content, misconceptions, a lack of interest, or a lack of personal experience, among others.

Following on from this idea, this study has shown that those initial teacher trainees who have learned through gamified activities in their school years and who have a higher level of knowledge about gamification will value more positively this type of strategy for the promotion of critical thinking and the principles of sustainability, equality, justice, and equity through controversial heritage; this, again, underlines the effectiveness of the implementation of training processes that use gamification at all educational stages, including teacher training. Furthermore, in order to explore these issues in greater depth, the future perspective of this work—and the limitations of this study—is to analyze, using this same instrument, the conceptions of initial teacher trainees regarding gamification and the teaching of controversial heritage in other countries, in order to be able to extrapolate the results to other contexts, as well as to study the responses of future teachers in greater depth using qualitative research instruments that allow us to understand the complexity of their conceptions.

**Author Contributions:** All the authors contributed equally to this study. All the authors wrote, reviewed, and commented on the manuscript. All authors have read and agreed to the published version of the manuscript.

**Funding:** This study is linked to the R&D+i project EPITEC2 "Controversial heritage for the eco-social formation of citizenship. An investigation of heritage education in formal education" (PID2020-116662GB-I00, funded by MCIN/AEI/10.13039/50100011033/) which has made its preparation possible. The first author is the beneficiary of a Teacher Training Grant (FPU20/01886), granted by the Ministry of Universities (Spain). The second author is a beneficiary of a FPI (PRE2021-097822), granted by MCIN/AEI/10.13039/501100011033 and the FSE+.

**Informed Consent Statement:** Written informed consent has been obtained from the participants to publish this paper.

**Data Availability Statement:** Not applicable.

**Acknowledgments:** This research has the support of the Research Group DESYM (HUM-168 of the PAIDI) and the Research Center on Contemporary Thought and Innovation for Social Development (COIDESO) of the University of Huelva, and the Red14: Network of Research in Social Sciences Teaching (RED2018-102336-T, granted by MCIN/AEI/10.13039/501100011033).

**Conflicts of Interest:** The authors declare no conflict of interest.

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
