# Peer review of "Gamification and Controversial Heritage: Trainee Teachers’ Conceptions"

_sustainability, doi:10.3390/su15108051_

Round 1
Reviewer 1 Report
The title and the aim of the paper are clear. The abstract is concise and informative in the same time. Reference list is quite relevant, recent and referenced correctly. The reference list can be improved as both topics of gamification and controversial heritage are covered in many academic journals and even books.
The Introduction is pertinent, with a highly organized structure. Consequently, the suggestion to enhance the reference list should be regarded as a minor revision recommendation.
The objectives of the research are very clearly defined. The Table 1 is very useful and explanatory.
The authors stated that results of the study “Ferreras-Listán, M., Pineda-Alfonso, J. A., & Hunt-Gómez, C. I. (2020). Heritage education as a tool for creating critical citizens: Analysis of conceptions of teachers in training. In Handbook of Research on Citizenship and Heritage Education (pp. 199-218). IGI Global” concluded that future secondary teachers showed a lack of relevance to heritage in their teaching and learning proposals. As the referenced source is relatively new and have only two citations, I suggest authors to name other studies arriving at the similar conclusion. I suggest that in the same paragraph (lines 528-532) to name few academic sources arriving at similar conclusions with your study results.
The study's outcomes are suitably displayed, and the conclusions are both relevant to the research objectives and presented into a clear way. In general, the paper receives a positive evaluation, and it merits publication, with only minimal revisions and the inclusion of some additional references needed.
Author Response
Dear reviewer,
Thank you for all your comments and suggestions. Here are the changes that we contemplated:
1) As the suggestion to improve the list of references has been considered as a minor revision recommendation, we have decided not to add to those already included for the following reasons: a) controversial heritages are a relatively new concept, whose origin, evolution and main references have already been mentioned in this work; b) from gamification we have chosen a wide range of works that serve as a reference for our study.
2) We have added the following quote "Estepa, J.; Ávila, R.; Ferreras, M. Primary and Secondary teachers' conceptions about heritage and heritage education: A comparative analysis. Teaching and Teacher Education, 2008, 24(8), 2095-2107. https://doi.org/10.1016/j.tate.2008.02.017" to support what we highlight from the study by Ferreras et al. (2020). We have also applied their suggestion about naming an academic source with similar conclusions to those of our study, adding the citation "Sampedro-Martín, S. Teachers' conceptions of role-playing in heritage teaching. A study in primary schools in Huelva. In La formación del profesorado en Didáctica de las Ciencias Sociales en el ámbito Iberoamericano; Sánchez Fuster, M.D.C., Campillo Ferrer, J.M., Vivas Moreno, V., Eds.; Editum: Murcia, Spain; 2021; pp. 229-241. https://doi.org/10.6018/editum.2919".
We would like to thank you again for your review.
Reviewer 2 Report
The article is convincingly written and reports findings that seem to be informative for readers.
However, the concept of heritage education may not be well known to readers and at first, the article seems closed; (See Umberto Eco for a definition). Page 2 of 21 is a closed text, the audience needs to be familiar with the concept shared or proposed by the author(s).
The abstract needs to include the final number of participants as well as (M=?;F=?), such information establishes validity of the research and its outcome.
The research questions are presented in page 5/21 and seem relevant to the research objectives. The discussion section addresses the research questions.
4.3 is an honest account of findings which aligns with other similar research. Just a note: an escape room is a phase and may not last as a teaching approach/method.
There is no limitation specifically stated (see 625~627).
The challenge with this article is that it does provide a new perspective for appropriate pre-service teacher training, and it does not specifically targets a wide international audience; how does the research informs US preservice teachers in terms of their own heritage awareness raising?
Author Response
Dear reviewer,
Thank you for all your comments and suggestions. Here are the changes that we contemplated:
1) We have added some references to clarifying that the heritage education is a long-standing concept developed in numerous publications, including one from this same research journal.
2) We have included the final number of participants in the abstract.
3) We have included the research questions right after the objectives, as you have recommended.
4) We have added the research questions immediately following the objectives.
5) We draw on other authors research to consider that escape room is a strategy, a gamified activity, and that is how we define it (Borrego et al., 2017; Eukel et al., 2017; Nicholson, 2018; Duggins, 2019; Sierra and Fernández, 2019; Moreno-Fernández et al., 2020; Veldkamp et al., 2020; Sampedro-Martín, 2023).
6) We have added some limitations of the study.
7) This study is aimed at teacher educators to start using such strategies, as it highlights the need to introduce gamification and controversial heritages in initial teacher education. Therefore, this study can be used in all initial teacher education processes in any country. Although the data are not directly extrapolatable, the research is replicable and the results are relevant to any other similar research that may be undertaken. Nevertheless, the heritage awareness raising of pre-service teachers is not the main focus of the article, but rather the objective is the relationship between gamification as a methodological strategy for the treatment of controversial heritages in initial teacher training.
We would like to thank you again for your review.
Reviewer 3 Report
The paper is well-written and sound. However, the following points should be rethought:
1) line 112: I cannot see which is the “key aspect” you mention.
2) line 244: must it be HE?
3) Table 2: Questions 1, 2, 3 and 4 refer to “the following statements, issues, heritage elements and group of items which may be controversial’. Please, write all of these elements somewhere, or, if they are written, explain clearly where.
4) Table 3: Please, review if the numbers of questions are rightly written.
5) lines 356-357: Please, review if it is correctly written
6) In pages 11, 12 and so on, \alpha-error of 0.000 is written. Please, write an \alpha-error less than 0.001.
7) line 530: some 50%? What does it mean?
8) line 559: I think that the expression “depersonalization of students“ is not what you want to express, is it?
Author Response
Dear reviewer,
Thank you for all your comments and suggestions. Here are the changes that we contemplated:
1) The “key aspect” is the initial teacher training. We have changed the order of the sentence to clarify what we are referring to.
2) Applied.
3) Applied.
4) The number of questions are rightly written. However, we have noted below the table that questions 3 and 9 do not include quantitative variables because they are different questions as we have explained above Table 2.
5) We consider that this sentence is correct, but if you notice a specific mistake, please note us.
6) Applied.
7) We have removed “some”.
8) That expression is exactly what we want to express, because the authors we have cited use the term “depersonalisation of students”.
We would like to thank you again for your review.